# AUDITING DATA CONTROLLER COMPLIANCE WITH DATA WITHDRAWAL

## ABSTRACT

We study auditing *total* data withdrawal, the case in which a user requests the exclusion of their data from both the training *and* test data for some machine learning task. This approach is motivated by the need for comprehensive compliance with data privacy regulations and legal frameworks around the world. We conceptualize the task of auditing total data withdrawal as an optimization problem. Compliance verification is conducted under mild assumptions using a dedicated verification algorithm. We then evaluate this formulation over various datasets, architectures, and verification hyperparameters. Our verification algorithm serves as a tool for regulators to ensure *auditable compliance* and provides enhanced privacy guarantees for users.

## 1 INTRODUCTION

Tech companies are increasingly seeking user data to enhance the training of their large-scale machine learning (ML) models. However, users frequently wish to impose limitations on the use of their data, particularly when it involves sensitive personal information. In response, governments and institutions have implemented policies to regulate data usage. A notable example is the European Union's General Data Protection Regulation (GDPR) (European Parliament & Council of the European Union, 2016a), which outlines strict restrictions on how tech companies, referred to as *data controllers*, can utilize user data.

The *right to be forgotten* mandates that a data controller must delete a user's data upon request. In other words, the data controller must not only delete the user's data but also remove the effect of the user's data on trained models (Federal Trade Commission, 2022). In response, the high cost of retraining models from scratch has led to the development of machine unlearning (Bourtoule et al., 2021). Unlearning enables data controllers to remove user data effects without retraining models from scratch. Furthermore, methods have been proposed to verify that machine unlearning has been properly executed (Sommer et al., 2022). However, unlearning *only* addresses data deletion during the training phase and does not fully comply with GDPR requirements. Thus, we say that unlearning implements *partial* data withdrawal over the training data.

Instead, *the right to object* states that a user can completely deny access to their data at *any stage of data processing*. That is, it is possible that the user data cannot be used at either train or test time. This sentiment extends beyond the GDPR to other policies for sensitive user data like the Healthcare Insurance and Portability Act (HIPAA) (Gostin, 2001) and is consistent with recent legal frameworks encouraging data transparency (California State Legislature, 2024).

In this work, we study *total* data withdrawal. Total data withdrawal refers to the user requesting complete exclusion of their data from all stages of data processing, including both training and testing phases. This ensures that the user's data is neither used to train the model nor used to make predictions. In particular, we study formal methods to audit data controller compliance with total data withdrawal. Verification methods are encouraged by policies like the GDPR (European Parliament & Council of the European Union, 2016b) and help establish trust by allowing users to audit compliance. Recent violations of user trust have damaged public opinion about data controllers and their data handling practices (U.S. SEC, 2019). Furthermore, lawsuits have been levied against data controllers for violating the right to object (O'Caroll, 2022). At scale, if data controllers do not prioritize compliance, this could be financially dangerous. Thus, data controllers are incentivized to act

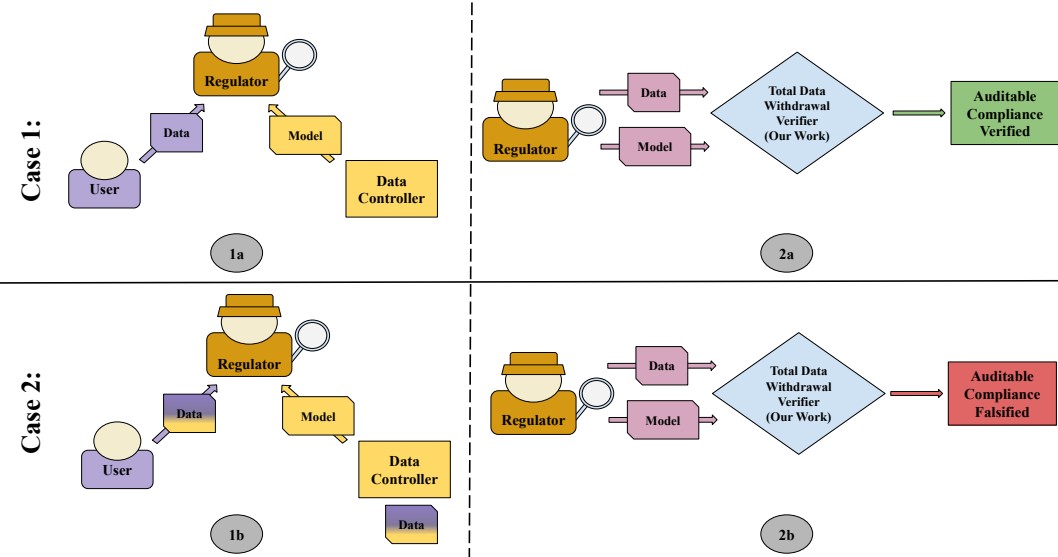

Figure 1: Demonstration of auditable compliance in two cases: when the data controller complies and when they do not. In both cases, the independent regulator collects the data from the user and the model from the data controller. Then, the regulator aims to assess whether the data controller retained the user's data for any purpose or not through the total data withdrawal verifier. In *case 1*, the regulator verifies auditable compliance. On the contrary, in *case 2*, the regulator falsifies. This implies that the data controller either used the data or failed to comply auditably.

transparently. Therefore, data controller compliance with total data withdrawal must be verifiable. We call this ***auditable compliance***.

To put this into practice, we study compliance with total data withdrawal audited by an unbiased third party regulator with access to the model and user data. We outline our setting in Fig. 1. This leads us to the following research question:

*Can a regulator audit data controller compliance with total data withdrawal?*

We answer affirmatively. Our contributions can be summarized as follows:

- We draw attention to total data withdrawal in machine learning, highlighting its importance for compliance with the law.
- We provide a necessary and sufficient condition for auditable compliance with total data withdrawal in Sec. 3.
- We manifest the first method to verify compliance with total data withdrawal in Sec. 4.
- We provide experiments evaluating verification performance over various models and image datasets in Sec. 5.

We make experiment code publicly available at `https://anonymous.4open.science/r/ICLR_Total_Data_Withdrawal/README.md`.

## 2 BACKGROUND AND RELATED WORK

To our knowledge, no previous work provably verifies compliance with total withdrawal. However, we review related work in adjacent areas. We particularly focus on machine unlearning and its verification methods (Sekhari et al., 2021), which, although different in goal, share similarities with our approach. Additionally, we discuss robustness verification (Salman et al., 2019), which we adapt to implement our verification framework. Furthermore, we draw connections between our work and broader concepts in data privacy.

**Machine Unlearning:** Machine unlearning is inspired by the GDPR's right to be forgotten, mandating that data controllers delete user data upon request (European Parliament & Council of the European Union, 2016a). In practice, data controllers must remove user data and its effects from trained models and algorithms. A naive approach is to retrain the model from scratch without the user's data. However, this is prohibitively expensive for large models (Sekhari et al., 2021). Thus, various methods have been developed to comply without retraining from scratch, including resource-efficient retraining (Bourtoule et al., 2021), differentially private unlearning (Sekhari et al., 2021), and finetuning through student-teacher knowledge distillation (Kurmanji et al., 2024).

However, unlearning does not ensure compliance at test time. To clarify, unlearning ensures *partial* data withdrawal by removing user data from the training set only. We propose *total* data withdrawal, excluding user data from all stages of processing, including training and testing. This approach provides more comprehensive compliance with regulations like the GDPR.

**Unlearning Verification and Auditing Compliance:** With growing interest in unlearning, verification methods have been increasingly studied (Wang et al., 2024). These methods involve hypothesis testing (Sommer et al., 2022), federated learning (Gao et al., 2024), data poisoning (Xu et al., 2024), measuring gradient changes (Zhou et al., 2024), and cryptographic primitives (Eisenhofer et al., 2022).

These methods audit compliance with partial data withdrawal, focusing on the training set. However, policies like the right to object require data controllers to exclude user data from all sources, including test data. Unlearning verification overlooks this while our method addresses it. Furthermore, current unlearning verification methods can be circumvented (Zhang et al., 2024) and thus may not guarantee data withdrawal.

Additionally, membership inference attacks (MIAs) provide an empirical notion of unlearning verification. We elaborate more on the differences between our work and MIAs in Appendix C.

More generally, our setting differs from typical compliance issues in machine learning, such as privacy-preservation (Steinke et al., 2024) or fairness (Yan & Zhang, 2022).

**Robustness Verification:** Our problem is loosely related to robustness verification (Salman et al., 2019), ensuring a neural network classifies all instances in an $\epsilon$-ball consistently. In doing so, one verifies that their model is robust to adversarial attacks within that $\epsilon$-ball (Goodfellow et al., 2014).

Beyond adversarial robustness, verification methods like this have been applied to power systems, aviation, and other fields (Bak et al., 2021). To the authors' knowledge, legal compliance has not been studied in this context.

**Data Privacy:** Due to the risk of data leakage from large information systems (Al-Rubaie & Chang, 2019), data privacy is a critical paradigm within computer science. Often, large data controllers such as social media companies use end-to-end encryption (Perrin & Marlinspike, 2016) to ensure privacy for their users. Since a data controller cannot use the data, they have no incentive to keep the data or violate privacy. This is an implicit premise of end-to-end encryption. Our formulation is posed similarly. It verifies that the data controller cannot use the data, and therefore has no incentive to keep the data or violate the law.

In the context of machine learning, data privacy is ubiquitous as well (Fredrikson et al., 2015). Approaches to privacy-preserving machine learning include differential privacy (Dwork et al., 2006) and homomorphic encryption (Brakerski et al., 2014). In some settings, our formulation can also verify privacy-preservation. Specifically, for identification tasks like facial recognition, our algorithm can accurately verify that a user cannot be identified. Thus, our work can also be thought of as a novel data privacy verification method. We leave this to future work.

# 3   A NECESSARY AND SUFFICIENT CONDITION FOR AUDITABLE COMPLIANCE

In what follows, we provide a verifiable condition for auditable compliance with total data withdrawal.

## 3.1 NOTATION

Bolded lowercase letters $x$ represent vectors and unbolded lower case letters $x$ represent scalars. $\mathcal{X}$ and $\mathcal{Y}$ represent the sample and label spaces, respectively. Functions are denoted $f : X \to Y$ where $X$ and $Y$ are sets. $f \circ g$ denotes the composition of functions $f$ and $g$. Subscripts and superscripts are used on sets differentiate sets which are similar in nature but different in content. $X \times Y$ denotes the Cartesian product of sets $X$ and $Y$. We use the shorthand $[n] = \{1, ..., n\}$ for positive integer $n$. $\min_{x \in X} (\max_{x \in X})$ denote the minimum (maximum) value of set $X$. For a neural network $f : \mathbb{R}^d \to \mathbb{R}^o$ the $i$th output logit for $x \in \mathbb{R}^d$ is $f(x)_i$. Thus, the class assigned by $f$ to $x$ is $\max_{i \in [o]} f(x)_i$. Refer to Appendix F for a keyword and symbol table.

## 3.2 PROBLEM SETTING

As a reminder, we aim to verify compliance with total data withdrawal for a task $T$. To that end, we define total data withdrawal as follows:

**Definition:** *Total data withdrawal is when a user requests that their data instance is neither used to train a machine learning model for task $T$ nor used for task $T$ itself at test time.*

This means we should present a condition that enables a regulator to confirm that the data controller has not used the user's data in either the train or test set. To clarify, it is insufficient for a regulator to verify unlearning alone. Unlearning verification only ensures partial data withdrawal during training and offers no guarantees at test time. Importantly, verifying data deletion from a provided test set is insufficient to ensure compliance at test time; an adversarial data controller could hide the true test set, and thus circumvent the law.

To establish a reliable condition, we let a regulator verify that the data controller misclassifies the user's instance. We also consider that the data controller might adversarially perturb the instance to ensure correct classification, even if the original instance is misclassified (Goodfellow et al., 2014). This altered instance could be used for task $T$, circumventing the law. To prevent this, we verify that the data controller misclassifies all instances within a closed ball around the original instance.

We formally demonstrate that this condition is both necessary and sufficient for verification. It is important to note that while this condition is not required for general compliance, it is essential for auditable compliance. A data controller can technically comply by simply deleting the data from both the training and test sets, yet still classify these instances correctly. However, such compliance cannot be audited, which undermines transparency. In our setting, transparency is crucial, and thus, satisfying this condition is necessary. Even if the data controller is trustworthy, they cannot achieve transparency without meeting this condition.

## 3.3 FORMALIZATION

Let $\mathcal{X} \subset [0,1]^d$ be our sample space and $\mathcal{Y} \subset \mathbb{R}^o$ be our label space. Suppose a data controller has a trained model $f_0 : [0,1]^d \to \mathbb{R}^o$ for use in downstream task $T$ with training data $S_{\text{tr}} = \{(x_i, y_i)\}_{i=1}^{N_{\text{tr}}} = X_{\text{tr}} \times Y_{\text{tr}}$ and test data $S_{\text{te}} = \{(x_i, y_i)\}_{i=1}^{N_{\text{te}}} = X_{\text{te}} \times Y_{\text{te}}$. Suppose a user $u$ totally withdraws their data instance $x_u$ for use in task $T$ where the data controller must classify $x_u$ correctly. Our task is to verify that neither $x_u$ nor any adversarially altered version of $x_u$ is used in either $S_{\text{tr}}$ or $S_{\text{te}}$ for task $T$. Without loss of generality, suppose $x_u$ is classified by $f_0$ as class $t$, i.e. $\max_{i \in [o]} f_0(x_u)_i = t$. Let $\bar{B}_\delta(x_u) = \{x \in [0,1]^d : ||x - x_u||_\infty \leq \delta\}$ be the closed $\delta$-ball around the data instance $x_u$. Let $f_m$ be a potentially altered version of $f_0$ used by the data controller for $T$ in place of $f_0$ after user $u$ invokes total data withdrawal.

Before we build a rigorous verification framework, the following assumptions are necessary:

**Assumption 1** $|\mathcal{Y}| \geq 3$, *i.e. we have three or more classes.*

**Assumption 2** $\exists \epsilon > 0$ *such that* $\forall x \in X_{tr}, X_{te}, ||x - x_u||_\infty > \epsilon$, *i.e. all instances are at least $\epsilon$ away from the object set instance. We let $\delta \leq \epsilon$ be the data controller's adversarial budget, i.e. how much they can perturb the original $x_u$.*

**Assumption 3** *The training loss of $f_m$ goes to 0.*

**Remarks:** These assumptions are realistic for popular modern machine learning tasks and datasets. We make this concrete in Appendix A. In Appendix A, we also provide more details on why these assumptions are necessary, the limitations that arise because of them, and how our formulation changes when they are relaxed.

We argue below that the data controller has auditably complied with total data withdrawal with respect to task $T$ if and only if the data controller misclassifies everything in $\bar{B}_\delta(\boldsymbol{x}_u)$, i.e. $\forall \boldsymbol{x} \in \bar{B}_\delta(\boldsymbol{x}_u)$, $\max_{i \in [o]} f_m(\boldsymbol{x})_i \neq t$:

( $\impliedby$ ) Suppose $f_m$ misclassifies all instances in $\bar{B}_\delta(\boldsymbol{x}_u)$. By Assumption 1, the data controller cannot identify $t$ by taking complements of $\max_{i \in [o]} f_m(\boldsymbol{x})_i$ for any $\boldsymbol{x} \in \bar{B}_\delta(\boldsymbol{x}_u)$. Thus, the data controller cannot use any $\boldsymbol{x} \in \bar{B}_\delta(\boldsymbol{x}_u)$ for task $T$. The data controller has no incentive to use any $\boldsymbol{x} \in \bar{B}_\delta(\boldsymbol{x}_u)$ for validation either, as it would confound generalization error calculation. Finally, by Assumption 3 the training loss of $f_m$ went to 0, implying that no instances in the training data are misclassified. That is, the data controller could not have used any $\boldsymbol{x} \in \bar{B}_\delta(\boldsymbol{x}_u)$ in the training data. Therefore, a regulator can verify data controller compliance.

( $\implies$ ) Suppose there exists a $\boldsymbol{x} \in \bar{B}_\delta(\boldsymbol{x}_u)$ such that $f_m$ classifies it correctly, i.e. $\exists \boldsymbol{x} \in \bar{B}_\delta(\boldsymbol{x}_u)$ s.t. $\max_{i \in [o]} f_m(\boldsymbol{x})_i = t$. By Assumption 2, $\boldsymbol{x}$ is either $\boldsymbol{x}_u$ or an adversarially perturbed version of $\boldsymbol{x}_u$, as all other instances are outside of $\bar{B}_\delta(\boldsymbol{x}_u)$. Thus, the data controller could use $\boldsymbol{x}$ for task $T$, violating total data withdrawal. It is not possible at this stage to verify whether or not the data controller will. As such, compliance is not verifiable. By contrapositive, a regulator must audit this condition to verify compliance.

## 4 VERIFICATION FORMULATION

We now introduce the verification formulation. Our formulation provides a *deterministic guarantee* for auditable compliance.

### 4.1 GENERAL FRAMEWORK

As discussed in Sec. 3, we must verify that every instance $\boldsymbol{x} \in \bar{B}_\delta(\boldsymbol{x}_u)$ is misclassified. Put differently, we must verify that $\forall \boldsymbol{x} \in \bar{B}_\delta(\boldsymbol{x}_u)$, $\exists \gamma \neq t$ s.t. $f_m(\boldsymbol{x})_\gamma > f_m(\boldsymbol{x})_t$ where $t = \max_{i \in [o]} f_0(\boldsymbol{x}_u)_i$, i.e. we misclassify everything in the $\delta$-ball where $f_0$ gives us our true class[1]. Rephrasing, $f_m(\boldsymbol{x})_\gamma > f_m(\boldsymbol{x})_t \iff f_m(\boldsymbol{x})_\gamma - f_m(\boldsymbol{x})_t > 0$. Therefore, it suffices to verify that the worst-case $\boldsymbol{x} \in \bar{B}_\delta(\boldsymbol{x}_u)$ satisfies this inequality.

We rewrite this as the following optimization problem:

$$v^* = \min_{\boldsymbol{x} \in \bar{B}_\delta(\boldsymbol{x}_u), \gamma \neq t} f_m(\boldsymbol{x})_\gamma - f_m(\boldsymbol{x})_t, \;\; \text{s.t.} \;\; t = \max_{i \in [o]} f_0(\boldsymbol{x}_u)_i \;\; \text{satisfies} \;\; v^* > 0. \quad (1)$$

Verifying compliance with total data withdrawal reduces to finding an efficient way to solve this optimization problem.

### 4.2 SOUND AND COMPLETE VERIFICATION

There are two main properties a verification algorithm satisfies: *soundness* and *completeness*. An algorithm is *sound* (*complete*) if every time it verifies (falsifies) a property, the answer is guaranteed to be correct (Liu et al., 2021). In our context, soundness means that whenever we verify, the data controller has auditably complied, and completeness means that whenever we falsify, the data controller has not auditably complied.

A verification algorithm is considered sound if it correctly verifies a property and complete if it correctly falsifies a property. In practical terms, for verification methods based on optimization such

---

[1]In order to verify misclassification, the regulator must know the true class. While we consider $t = \max_{i \in [o]} f_0(\boldsymbol{x}_u)_i$ as the true class, obtained from the data controller's model $f_0$, the regulator is also able to obtain the true label of $\boldsymbol{x}_u$ from user $u$ before verification. This is consistent with solutions posed in robustness verification (Liu et al., 2021).

as Eq. (1), achieving the global minimum of the objective function ensures both sound and complete verification. Our focus is on ensuring both soundness and completeness. Soundness assures users that their data is secure, while completeness ensures that data controllers can trust third-party audits for compliance.

Given that $f_m$ is a ReLU network, achieving both soundness and completeness in the context of Eq. (1) necessitates solving a linear optimization problem with linear constraints (Boyd & Vandenberghe, 2004). Drawing inspiration from robustness verification literature, we implement the state-of-the-art verification approach, $\beta$-CROWN BaB (Wang et al., 2021), to the novel task of verifying total data withdrawal. The following analysis shows how this method meets our criteria for soundness and completeness.

**Proposition 1** *(Theorem 3.3 of (Wang et al., 2021)) Suppose we are verifying $\forall x \in \bar{B}_\delta(x_u)$, $f(x) > 0$ for some $f : \mathbb{R}^d \to \mathbb{R}$ with ReLU activation functions $\sigma(x) = \max\{0, x\}$. Then, $\beta$-CROWN BaB is sound and complete.*

**Proposition 2** *Suppose that $f_m$ is a neural network with ReLU activation functions $\sigma(x) = \max\{0, x\}$ and $t = \max_{i \in [o]} f_0(x)_i$ for some model $f_0$. Then, there exists a $g : \mathbb{R}^o \to \mathbb{R}$ such that Eq. (1) is satisfied if and only if $\forall x \in \bar{B}_\delta(x_u)$, $h(x) > 0$ where $h = g \circ f_m$ is satisfied.*

For brevity, the proof of proposition 2 and our choice of $g$ are given in Appendix D.

**Corollary** *The studied verification algorithm is sound and complete for our formulation.*

**Remarks:** Not all sound and complete verification algorithms are adaptable to our setting. For example, although $\alpha$-convexification verification (Abad Rocamora et al., 2022) can be extended to twice-differentiable classifiers, it cannot solve our objective correctly. This is because algorithms that rely on verifying a single class $\gamma$ at a time–that is, finding the minima across only $\bar{B}_\delta(x_u)$ and not all $\gamma$–cannot be used. If we fix an arbitrary $\gamma$ and falsify because the true class's logit is greater than $\gamma$'s logit for some instance in $\bar{B}_\delta(x_u)$, we could falsify incorrectly. Another class's logit might be greater than the true class logit for that instance. Thus, we must find the minima across all classes that are not $t$.

Finally, in practice, a data controller would likely verify compliance for multiple instances. Consider the "object set" $X_o \subset X_{tr} \cup X_{te}$. Let $X_o^{adv} = \bigcup_{i=1}^{|X_o|} \bar{B}_\delta(x_i)$ where $x_i \in X_o$, i.e. the set of all closed $\delta$-balls of the instances in the object set. Thus, our problem involves verifying Eq. (1) for all $x \in X_o^{adv}$. Notably, we consider $\epsilon$ to satisfy $\forall x_n, x_m \in X_{tr}, X_{te}, \|x_n - x_m\|_\infty > \epsilon$ for $n \neq m$, i.e. all data instances are at least $\epsilon$ apart. To clarify, our algorithm works in this case because this requirement implies Assumption 2. We discuss this assumption in Appendix A.

## 5 EXPERIMENTS

We present numerical evidence demonstrating the effectiveness of our algorithm in verifying total data withdrawal. Our approach is straightforward for identification tasks, such as face recognition, where removing the identifying label of $x_u$ is sufficient to induce misclassification. However, inducing misclassification without merely deleting the label is more complex. Therefore, to extend our approach beyond identification tasks, we have designed custom benchmarks.

**Benchmarks**: We construct two primary benchmarks to make a clear distinction between the object set and the rest of the data. We use the popular datasets MNIST (Lecun et al., 1998), FashionMNIST (FMNIST) (Xiao et al., 2017), Kuzushiji-MNIST (KMNIST) (Clanuwat et al., 2018), and CIFAR-10 (Krizhevsky et al., 2009) as the starting datasets. These widely used datasets represent a diverse set of challenges for our verification method.

In the first benchmark, which is applied to MNIST , KMNIST , FMNIST , we transform the base dataset as follows:

i) We tint all $x \in \mathcal{X} \setminus X_o$ green;

ii) We tint all $x \in X_o$ red, rotate them 15 degrees, and add Gaussian noise pixel-by-pixel sampled from $\mathcal{N}(0, 0.01)$.

To make this clear, we illustrate our transformation in Fig. 3. Additionally, we include ablation studies on why simpler transformations do not suffice in Appendix B.

Then, for our second benchmark, we downsample CIFAR10 by 5 classes to obtain "CIFAR 5". We use this throughout as a more complex, modern benchmark. Since this dataset is more nuanced, we need a special transformation to generate our custom dataset for verification. More details about this transformation are included in Appendix E.

**Setup:** We utilize a ReLU-MLP architecture (named as '2L ReLU') and ResNet18 (He et al., 2016). More information about our datasets, architectures, choices of $\delta$, and other experimental details are given in Appendix E. We use $\alpha\beta$-CROWN as our verification backend with further details included in Appendix E.1.

**Paradigms:** Our two core experimental paradigms are:

a) correctly *verifying* that a data controller has auditably complied when they *remove* object set instances from the test data.

b) correctly *falsifying* that a data controller has auditably complied when they *retain* object set instances in their training data.

In paradigm b), these object set instances are adversarially altered with a projected gradient descent (PGD) attack (Madry et al., 2019) with $\epsilon = \delta$, $\alpha = \frac{2}{255}$, and 50 restarts. We provide experiments for when they are not adversarially altered in Appendix B. We present results of test set experiments a) in Table 1 and train set experiments b) in Table S11. In our experiments, $f_0$ is trained and tested on the original dataset, while $f_m$ is trained and tested on the transformed dataset.

Additionally, there is no theoretical difference between paradigm a) and correctly verifying that a data controller has auditably complied for the train set. In particular, in both cases, to return the correct answer we must verify $\delta$-ball misclassification. However, empirically, these two paradigms can differ slightly due to distribution shift between train and test data. We test this case in Appendix B, finding that we obtain similar results as in paradigm a).

**Baselines:** To provide a baseline for experimental paradigm a), we verify that there does not exist a nearest neighbor (Fix, 1985) in the test set's embedding space for our object set instances. We take this to be an *upper* bound on the number of instances we can verify. This provides a nontrivial solution to our problem that does not require the data controller to share the entire dataset. For experimental paradigm (b), we compare our method to unlearning verification algorithm Athena (Sommer et al., 2022). We also implement the same upper bound method, albeit providing us with the number of instances we can falsify. More details on these baselines are provided in Appendix E.3. We perform additional ablation studies in Appendix B, examining edge cases for our verification algorithm, alternative architectures, different choices of verification hyperparameters, and other settings. Code for our experiments is available at `https://anonymous.4open.science/r/ICLR_Total_Data_Withdrawal/README.md`.

**Results:** As shown in Table 1, our verification formulation works correctly with occasional failures due to limitations in our custom dataset generation process. Additionally, as shown in Table 2, we falsify correctly so long as our models reach approximately 0 train error over the transformed dataset. From our CIFAR5 results, it is clear why Assumption 3 is necessary; without the train loss going to 0, our correctness is limited. Additionally, Athena when used in our setting often fails when verifying perturbed instances. While Athena in its own, different setting generally works well, it requires the strong assumption that the dataset is poisoned with backdoor attacks beforehand. Furthermore, this version of Athena also breaks for more complex datasets like CIFAR5. This demonstrates the affordances of our method. Experiments presenting what happens in paradigm b) when we verify over the original object set instances are presented in Appendix B.

Additionally, as pictured in Fig. 2, we find that as $\delta$ grows exponentially, auditable compliance becomes exponentially harder to verify. This is because as $\delta$ grows, so does the size of the $\delta$-ball and hence the number of domains to verify. However, the relationship between falsification and $\delta$ is less clear; MNIST remains relatively stagnant, while FMNIST becomes much harder to falsify as $\delta$ increases.

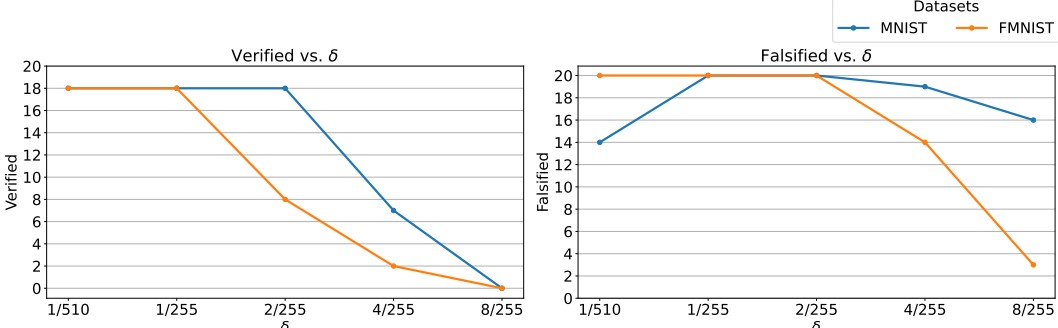

Figure 2: Ablation study on the correct verification (falsification) rate as $\delta$ grows. Instances are verified (falsified) over a 2L ReLU trained over MNIST (blue) or FMNIST (orange). As $\delta$ increases, verification and falsification both tend to get more challenging.

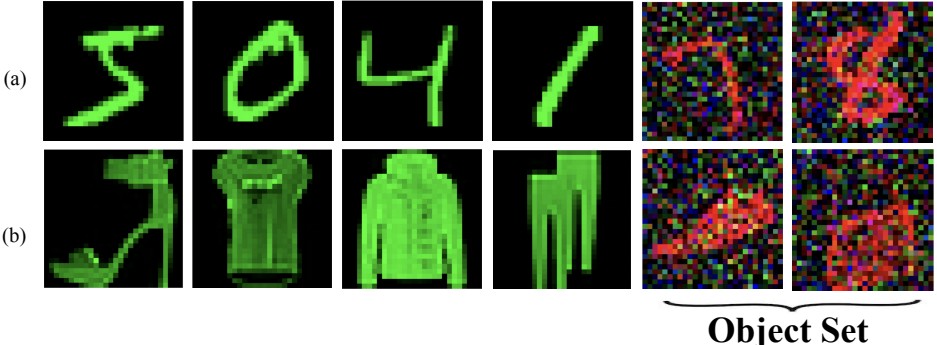

**Object Set**

Figure 3: We demonstrate our transformation on a) MNIST and b) FMNIST. Green instances are outside of the object set while red and rotated instances with noise applied are in the object set.

## 6 DISCUSSION

In this work, we examine total data withdrawal, where the data controller is prohibited from using a user's instance during both training and testing phases. To enhance trust between users and data controllers, it is essential that an independent regulator can audit the data controller for legal compliance. Accordingly, we propose a novel method for auditing compliance. Our sound and complete algorithm ensures that the data controller adheres to total data withdrawal requirements. Our method enables efficient verification for identification tasks, where deleting the user's label induces misclassification. We provide both formal and empirical evidence demonstrating that our approach accurately verifies and falsifies compliance.

Table 1: Ensuring that we verify correctly when the data controller excludes the object set from the test set. That is, the more instances we verify with respect to the upper bound, the better our performance. Notice that we perform relatively well throughout with slight drops for KMNIST and CIFAR5.

| Dataset | Network | Verified | Upper Bound | $f_m$ Test Accuracy |
|---------|---------|----------|-------------|---------------------|
| MNIST   | 2L ReLU | 18       | 20          | 98%                 |
| FMNIST  | 2L ReLU | 18       | 20          | 86%                 |
| KMNIST  | 2L ReLU | 12       | 20          | 88%                 |
| CIFAR5  | ResNet  | 7        | 10          | 88%                 |

Table 2: Ensuring that we falsify correctly when the data controller includes the object set in the train set. That is, the more instances we falsify with respect to the upper bound, the better our performance. In general, high train accuracy leads to good performance across datasets. We denote by regular Athena the unlearning verification algorithm adapted to our setting, while Athena* requires stronger assumptions on the data modeling than our setting, but is added for thoroughness.

| Dataset | Method | Network | Falsified | Upper Bound | $f_m$ Train Accuracy |
|---------|--------|---------|-----------|-------------|----------------------|
| MNIST | Athena* | 2L ReLU | 0 | 20 | 100% |
|  | Athena | 2L ReLU | 0 | 20 | 100% |
|  | Ours | 2L ReLU | 20 | 20 | 100% |
| FMNIST | Athena* | 2L ReLU | 20 | 20 | 97% |
|  | Athena | 2L ReLU | 0 | 20 | 97% |
|  | Ours | 2L ReLU | 20 | 20 | 97% |
| KMNIST | Athena* | 2L ReLU | 20 | 20 | 100% |
|  | Athena | 2L ReLU | 20 | 20 | 100% |
|  | Ours | 2L ReLU | 20 | 20 | 100% |
| CIFAR5 | Athena* | ResNet | 0 | 10 | 87% |
|  | Athena | ResNet | 0 | 10 | 87% |
|  | Ours | ResNet | 3 | 10 | 87% |

**Limitations:** Our formulation is limited to the $\delta$-ball, multiclass classification settings, and $f_m$ which have approximately 0 training loss. Beyond the label deletion case, our method relies on the data controller's ability to induce misclassification of particular instances.

**Future Directions:** Firstly, to address our limitations, future directions include designing verification formulations for binary classification, regression tasks, $f_m$ that do not interpolate perfectly, and that handle adversarial changes to $\mathbf{x}_u$ independent of $\delta$. Additionally, data controllers need algorithms to efficiently obtain $f_m$ from $f_0$, i.e. inducing misclassification over $X_o^{\mathrm{adv}}$, without severely affecting the generalization of $f_m$.

Next, while neural network verification techniques are natural for our problem, probabilistic relaxations with hypothesis testing (Sommer et al., 2022) might improve resource-efficiency, extend to different architectures, and address our limitations.

Then, extending this method to diverse datasets across different modalities, like text, is an important challenge. Total data withdrawal can be invoked over non-image data as well. Verifying over text remains a difficult problem in verification more broadly (Casadio et al., 2024).

Finally, working with regulators to bring this method into practice is a worthwhile future direction as well.

## BROADER IMPACT

In this work, we tackle the important problem of auditing models to ensure they obey the right-to-object right of users. By advancing this area, we can positively impact society by ensuring data controllers (e.g., model providers) deploy models that only use appropriate data for training and testing.

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

## CONTENTS OF THE APPENDIX

The supplementary material is organized as follows:

- In Appendix A, we provide additional details on Assumption 1, Assumption 2, and Assumption 3. Specifically, we demonstrate that Assumption 1 and Assumption 2 hold over various popular datasets. We also validate the necessity of Assumption 3 and discuss how our formulation changes when assumptions are relaxed.

- In Appendix B, we present various ablation experiments on edge cases for verification, our transformation, and experiments over alternative architectures and settings. We provide a table of contents for these ablation studies at the beginning of the section as well.

- In Appendix C, we provide a background on membership inference attacks (MIAs) and detail how our work differs from them.

- In Appendix D we prove proposition 2 by expanding on our choice of $g$ required to make $\beta$-CROWN BaB sound and complete for our formulation.

- In Appendix E, we expand on experimental details that were omitted in Sec. 5 for brevity. First, in Appendix E.1, we provide the $\alpha\beta$-CROWN specifications we use for our experiments. Next, in Appendix E.2, we provide additional details on our network architectures and datasets used. Then, we provide details on our baselines in Appendix E.3. Finally, we present further experimental details, like the methods used to train our networks, for reproducibility in Appendix E.4.

- In Appendix F, we provide a table containing symbols and keywords used throughout the paper.

Table S3: Number of classes in common datasets.

| Dataset | Class Count |
|---|---|
| MNIST | 10 |
| FMNIST | 10 |
| KMNIST | 10 |
| CIFAR5 | 5 |
| CIFAR10 | 10 |
| CIFAR100 | 100 |
| ImageNet | 1000 |

Table S4: Minimum distance between two instances in the input and embedding space. Distances are given with respect to the $\ell_\infty$ norm. The embeddings are taken from the last layer of ResNet50 pre-trained over ImageNet. Our $\delta$ are always smaller than these.

| Dataset | Input Space | Embedding Space |
|---|---|---|
| MNIST | 0.41 | 0.66 |
| FMNIST | 0.33 | 0.81 |
| KMNIST | 0.88 | 0.94 |
| CIFAR5 | 0.33 | 1.1 |
| CIFAR10 | 0.27 | 0.98 |
| CIFAR100 | 0.20 | 0.90 |
| ImageNet | 0.20 | 0.92 |

## A   ASSUMPTIONS

To begin, we provide evidence that Assumption 1 holds for many datasets used in machine learning research, some of which are used in Sec. 5. From Table S3, it is clear that many datasets do not consider binary classification as their setting.

For Assumption 2, we consider it in its most general form where all instances in the train and test data are at least $\epsilon$ apart. We provide this $\epsilon$ for common datasets in both input and embedding spaces in Table S4. From Table S4, it is clear that $\epsilon$ is usually fairly large in both the input and embedding spaces, especially when compared to the adversarial budgets in the robustness literature Bai et al. (2021). Note that the $\delta$ used in Sec. 5 are always lower than the smallest $\epsilon$ for that dataset across the pixel and embedding spaces.

To elaborate on what occurs when our assumptions are relaxed, consider the case that Assumption 1 is relaxed. Then, the data controller can find the true class for $\boldsymbol{x}_u$ by taking complements of $f_m(\boldsymbol{x}_u)$ and thus use $\boldsymbol{x}_u$ for task $T$. Thus, our condition no longer becomes sufficient.

Then, consider what occurs when Assumption 2 is relaxed. Then, there might exist an instance $\boldsymbol{x}_k$ belonging to user $k$ in $\bar{B}_\delta(\boldsymbol{x}_u)$. This instance $\boldsymbol{x}_k$ may be classified as $t$, although all instances related to $\boldsymbol{x}_u$ in $\bar{B}_\delta(\boldsymbol{x}_u)$ are misclassified. Thus, we would be falsifying incorrectly. In particular, our condition no longer becomes necessary and our verification formulation incomplete by default.

From the CIFAR5 results in Table 2, it is clear that we do not falsify correctly if our train loss does not converge to 0 or approximately 0. This illustrates the necessity of Assumption 3. Thus, in the case that Assumption 3 is relaxed, our method verifies partial data withdrawal over the test set. Notably, neural networks satisfying Assumption 3 are often called interpolators and are of interest in deep learning theory (Taheri & Thrampoulidis, 2023).

## B   ABLATIONS

We conduct the following ablation studies:

Table S5: Ensuring that we verify correctly when the object set is a subset of the train set and the data controller complies. With both our method and Athena, we do so.

| Method | Dataset | Network | Verified | Upper Bound | $f_m$ Train Accuracy |
|--------|---------|---------|----------|-------------|----------------------|
| Ours | MNIST | 2L ReLU | 9 | 10 | 100% |
| Athena* | MNIST | 2L ReLU | 10 | 10 | 100% |

Table S6: Ensuring that we verify correctly when the object set is a subset of the test set when we apply different transformations. As one can see, removing any aspect of our default transformation results in poor performance.

| Dataset | Network | Transformation | Verified | Upper Bound |
|---------|---------|----------------|----------|-------------|
| **MNIST** | **2L ReLU** | **Tint, Rotate, G. Noise** | **9** | **10** |
| MNIST | ResNet | Tint, Rotate, G. Noise | 0 | 10 |
| MNIST | 2L ReLU | Tint, G. Noise | 8 | 10 |
| MNIST | 2L ReLU | Tint, Rotate | 2 | 10 |
| MNIST | 2L ReLU | Tint | 2 | 10 |
| MNIST | 2L ReLU | G. Noise | 1 | 10 |

1. We perform experiments in the case that $f_m$ is trained over the synthetic dataset *without* the object set instances in the training data, i.e. $f_m$ is trained over $X_{\text{tr}} \setminus X_{\text{o}}$. We do so in Appendix B.1. This demonstrates that we can accurately verify unlearning as well, while experiment paradigm b) in the main paper demonstrates that we can accurately falsify unlearning.

2. We demonstrate that our transformation, as it is above, is vital in Appendix B.2. We also demonstrate what happens when we apply our more advanced CIFAR5 transformation to paradigm b). This illustrates why we choose the synthetic data generating method we do. We also demonstrate that our choice of colors are immaterial, so long as they are visually distinct.

3. We present results for experiment paradigm b) on larger architectures and full datasets in Appendix B.3. This illustrates that, while we consider paradigm b) on limited datasets and architectures, our experiments extend to broader settings.

4. We demonstrate what happens when we conduct experiment paradigm b) on the original object set instances rather than the perturbed object set instances in Appendix B.4. This provides deeper insights into our method as used for classic unlearning verification.

## B.1 EXPERIMENTS ON VERIFYING UNLEARNING

Here, we study verifying object set instances from the training data when they are excluded from the train data for $f_m$. Since $f_m$ is compliant, we should verify in this case.

From Table S5, we see that we obtain similar results to when we verify over the test set. Note that the version of Athena displayed in this case is the one that requires poisoning.

## B.2 EXPERIMENTS ON THE NECESSITY OF OUR TRANSFORMATIONS

**Default Transformation:** Here, we study verification over the test set when we remove aspects of our transformation as shown in Fig. S4.

We began by simply adding Gaussian noise sampled from $\mathcal{N}(0, 0.01)$. We then tried adding on tinting the object set red and the rest of the dataset green. We additionally tried adding a 15 degree rotation. We see that this provided great results for the multilayer perceptron (MLP) in Table S6. However, from Table S6, we see that all three of tinting red, rotating, and adding Gaussian noise does not suffice for our miniature ResNet18. This motivates the use of our current transformation for

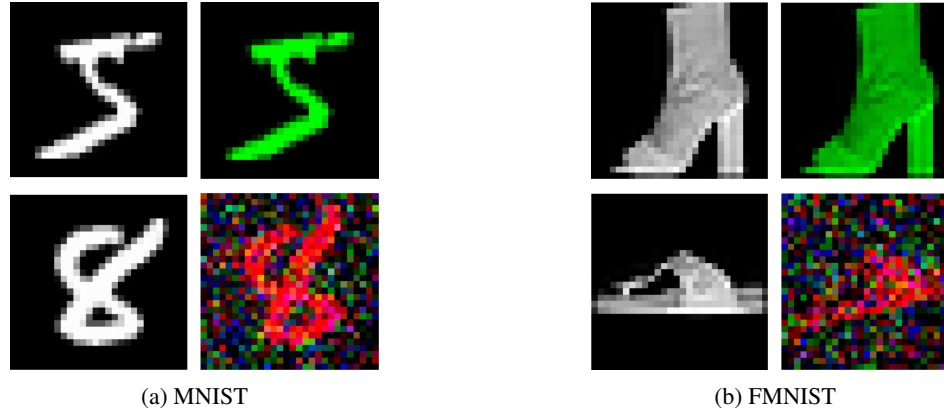

(a) MNIST               (b) FMNIST

Figure S4: Samples from our datasets and how they are created. This is demonstrated for MNIST (left) and FMNIST (right). We transform instances in the object set (e.g. the MNIST 8 instance) by tinting them red, rotating them by 15 degrees, and adding noise. To transform instances outside of the object set, (e.g. the MNIST 5 instance) we tint them green.

Table S7: Ensuring that we falsify correctly when the object set is the subset of the train set. Here, we mirror paradigm b). In this case, we use the CIFAR5 transformation for all experiments, which has no profound effect either way.

| Method | Dataset | Network | Falsified | Upper Bound | $f_m$ Train Accuracy |
|---|---|---|---|---|---|
| Ours | MNIST | 2L ReLU | 20 | 20 | 100% |
| Ours | FMNIST | 2L ReLU | 10 | 10 | 89% |
| Ours | KMNIST | 2L ReLU | 20 | 20 | 100% |
| Ours | CIFAR5 | ResNet | 3 | 5 | 87% |

smaller models, and using the CIFAR5 transformation detailed in Appendix E for larger architectures. In our main experiments, a ReLU MLP sufficed to verify well. Note that this transformation does not obfuscate the ground truth class in MNIST and its variants.

**CIFAR5 Transformation:** Here, we elaborate on why our CIFAR5 transformation is necessary for CIFAR5. Namely, the original, default perturbation used for MNIST in its variants obfuscates the ground truth classes in CIFAR5. This is due to the fact that CIFAR5 images are naturally colored. When conducting paradigm b) with the CIFAR5 transformation over FMNIST, KMNIST, and MNIST, as shown in Table S7, we see similar results as in Table 2.

**Color Transformation:** Here, we briefly show that, while the essential aspects of our transformation are important, some minor details are not. Specifically, we study how the colors red and green affect our transformation. We proceed by sampling two colors, sampling a real number in $[0, 1]$ uniformly at random for each channel. We demonstrate how these colors alter MNIST digits in Fig. S5

From Table S8 and Table S9, we see that we obtain similar results to when we use red and green. As such, the choice of colors in our transformation is unimportant, so long as they are visually distinct.

Table S8: Paradigm a) experiments with a different color transformation.

| Dataset | Network | Instance | Upper Bound |
|---|---|---|---|
| MNIST | 2L ReLU | 9 | 10 |

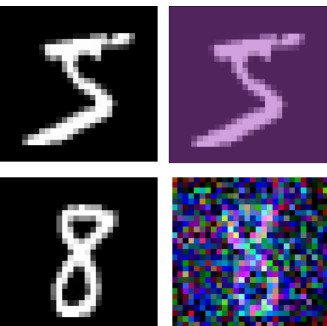

Figure S5: We consider a different transformation, where we tint instances outside of the object set (e.g. the 5 instance) a different color than green and instances within the object set (e.g. the 8 instance) a different color than red.

Table S9: Paradigm b) experiments with a different color transformation.

| Dataset | Network | Falsified | Upper Bound |
|---------|---------|-----------|-------------|
| MNIST | 2L ReLU | 10 | 10 |

## B.3 Experiments with Larger Architectures and Datasets

Here, we conduct experiments in paradigm b) for our method with various larger architectures and datasets. None of the datasets included in this section are downsampled. Note that due to limitations in $\beta$-CROWN, we cannot conduct experiments in paradigm a) with larger architectures than our ResNet. Extended verification algorithms to larger networks e.g. at the ImageNet scale remains an open problem Wang et al. (2021). Note that it suffices to falsify a perturbed object set instance for falsification, since this instance is in the $\delta$-ball.

## B.4 Experiments on Falsifying Unlearning over the Original Object Set

As demonstrated in Table S11, so long as we reach 0 or approximately 0 training error, we verify correctly. In these cases, Athena verifies well too; the main gap in Athena, and unlearning verification methods more broadly, that this work addresses is that they do not correctly falsify compliance when the data controller retains perturbed instances.

## C Membership Inference Attacks

Membership inference attacks (MIAs) (Shokri et al., 2017) provide a way to infer whether an instance is in the training data or not. Several approaches to MIAs include those based on confidence scores (Hayes et al., 2018), "shadow training" of models to classify membership (Shokri et al.,

Table S10: Ensuring that we falsify correctly when the object set is the subset of the train set. Notice that as models get significantly larger, since the train loss goes to 0 we are able to falsify with 100% accuracy. This further justifies our Assumption 3.

| Dataset | Network | Falsified | Upper Bound | $f_m$ Train Accuracy |
|---------|---------|-----------|-------------|----------------------|
| MNIST | CNN | 10 | 10 | 100% |
| FMNIST | CNN | 10 | 10 | 100% |
| CIFAR10 | Full ResNet | 20 | 20 | 91% |

Table S11: Ensuring that we falsify correctly when the object set is the subset of the train set, in particular when the data controller is still using the original object set. All datasets are downsampled MNIST and all models are 2L ReLU.

| Method | Falsified | Upper Bound | $f_m$ Train Accuracy |
|--------|-----------|-------------|----------------------|
| Ours | 9 | 10 | 100% |
| Athena* | 10 | 10 | 100% |

2017), and various metrics like rescaled loss (Yeom et al., 2018). Often, MIAs are used to evaluate the efficacy of unlearning methods (Kurmanji et al., 2024).

Notably, MIAs differ from our setting in two keys ways. First, MIAs do not offer formal guarantees about whether an instance belongs to the training data or not. Our work, and unlearning verification algorithms, aim to provide these formal guarantees. Furthermore, MIAs still only operationalize partial data withdrawal. To the authors' knowledge, there exist no MIAs that infer test set membership of an instance. As such, the goal of our work and the paradigm we operate in are both different.

## D    PROOF OF PROPOSITION 2

Here, we elaborate on our choice of $g : \mathbb{R}^o \to \mathbb{R}$ such that proposition 2 holds.

To begin our argument, suppose $f_m$ is a neural network with ReLU activation functions $\sigma(\boldsymbol{x}) = \max\{0, \boldsymbol{x}\}$ and that $t = \max_{i \in [o]} f_0(\boldsymbol{x})_i$ for some model $f_0$. We want to find a $g$ such that our formulation in Sec. 4 is equivalent to satisfying $\forall \boldsymbol{x} \in \bar{B}_\delta(\boldsymbol{x}_u), \ h(\boldsymbol{x}) > 0$ where $h = g \circ f_m$.

For $\boldsymbol{y} \in \mathbb{R}^o$, define $g(\boldsymbol{y})$ as

$$g(\boldsymbol{y}) = \begin{cases} 1 & ||\boldsymbol{y}||_\infty \neq t \\ -1 & ||\boldsymbol{y}||_\infty = t \end{cases}$$

Define $h = g \circ f_m$. We want to show that verifying Eq. (1) and verifying

$$\forall \boldsymbol{x} \in \bar{B}_\delta(\boldsymbol{x}_u), \ h(\boldsymbol{x}) > 0 \tag{2}$$

are equivalent. To show this, we demonstrate that we verify Eq. (2) if and only if we verify Eq. (1).

( $\implies$ ) Suppose we verify Eq. (2). Then, $\forall \boldsymbol{x} \in \bar{B}_\delta(\boldsymbol{x}_u), \ h(\boldsymbol{x}) > 0 \implies \forall \boldsymbol{x} \in \bar{B}_\delta(\boldsymbol{x}_u), \ g(f_m(\boldsymbol{x})) > 0 \implies \forall \boldsymbol{x} \in \bar{B}_\delta(\boldsymbol{x}_u), \max_{i \in [o]} f_m(\boldsymbol{x})_i \neq t \implies \forall \boldsymbol{x} \in \bar{B}_\delta(\boldsymbol{x}_u), \ \exists \gamma \neq t$ s.t. $f_m(\boldsymbol{x})_\gamma > f_m(\boldsymbol{x})_t \implies \forall \boldsymbol{x} \in \bar{B}_\delta(\boldsymbol{x}_u), \ \exists \gamma \neq t$ s.t. $f_m(\boldsymbol{x})_\gamma - f_m(\boldsymbol{x})_t > 0$, which implies that we verify Eq. (1).

( $\impliedby$ ) Supposes that we verify Eq. (2) falsifies. Then, $\exists \boldsymbol{x} \in \bar{B}_\delta(\boldsymbol{x}_u)$ s.t. $h(\boldsymbol{x}) \leq 0 \implies \exists \boldsymbol{x} \in \bar{B}_\delta(\boldsymbol{x}_u)$ s.t. $g(f_m(\boldsymbol{x})) \leq 0 \implies \exists \boldsymbol{x} \in \bar{B}_\delta(\boldsymbol{x}_u)$ s.t. $\max_{i \in [o]} f_m(\boldsymbol{x})_i = t \implies \exists \boldsymbol{x} \in \bar{B}_\delta(\boldsymbol{x}_u)$ s.t. $\forall \gamma \neq t \ f_m(\boldsymbol{x})_\gamma \leq f_m(\boldsymbol{x})_t \implies \exists \boldsymbol{x} \in \bar{B}_\delta(\boldsymbol{x}_u)$ s.t. $\forall \gamma \neq t \ f_m(\boldsymbol{x})_\gamma - f_m(\boldsymbol{x})_t \leq 0$, which implies that we falsify Eq. (1). Thus, by contrapositive, the conclusion follows.

In practice, $g$ can be implemented by obtaining $t$ beforehand and then iterating over all output logits of $f_m(\boldsymbol{x})$ to obtain $\max_{i \in [o]} f_m(\boldsymbol{x})_i$. Then, we can evaluate $g$ as it is defined.

## E    ADDITIONAL EXPERIMENT DETAILS

Here, we provide additional details for our experiments in Sec. 5. Specifically, we provide our specifications for our verification backend $\alpha\beta$-CROWN and details about the models and datasets we use. We also provide additional details about our baselines. Finally, we provide exact details on our main paper experiments for reproducibility.

### E.1  $\alpha\beta$-CROWN SPECIFICATIONS

To run $\alpha\beta$-CROWN, we consider the following:

- We generate a file custom.py to store our model definitions.

- For each object set instance, we generate a VNNLib file (Demarchi et al., 2023) bounding each pixel with respect to its $\ell_\infty$ bounds per $\bar{B}_\delta(x_u)$. We get the true class of $x_u$ beforehand and consider the output specifications in line with Appendix D.

- For the yaml file, we let the operations be deterministic, use customized model definitions in custom.py, and load in object set instances via a VNNLib CSV. We also set the batch size to 2048 and the maximum branch-and-bound iterations to 30 in order to preserve GPU memory. Finally, we set the operations to be deterministic for reproducibility. Otherwise, we use default arguments.

Since we use default arguments, the following occurs during verification:

1. We run a PGD attack on the object set instance, performing complete but not sound verification. In our case, we are trying to find an instance in $\bar{B}_\delta(x_u)$ that is classified correctly. If this fails, we proceed.

2. Then, we run CROWN, performing sound but incomplete verification. We do so as this tends to be faster than running $\beta$-CROWN alone. If this fails i.e. CROWN falsifies, we proceed.

3. Finally, we run $\beta$-CROWN, performing sound and complete verification.

### E.2  NETWORK AND DATASET DETAILS

**2L ReLU:**  A two-layer ReLU network with a hidden dimension of 100 neurons.

**ResNet:**  A ResNet18 with an initial convolutional layer (LeCun et al., 1995), batch normalization (Ioffe & Szegedy, 2015), then [2,2,2,2] residual blocks (He et al., 2015). This architecture is a smaller version of the original ResNet18. With respect to output convolution channels per stage of ResNet18, instead of [64,128,256,512] we have [2,4,8,8]. This architecture is, to the authors' knowledge, the largest model that can verify over CIFAR5 for $\beta$-CROWN without timing out or resulting in an out-of-memory error. We pull the architecture definition used in our code from the $\alpha\beta$-CROWN repository.

**CNN:**  A convolutional neural network composed of three identical blocks and a final fully connected linear layer. A block is composed of a ReLU layer, followed by a convolutional layer (LeCun et al., 1995), followed by batch normalization (Ioffe & Szegedy, 2015). Convolutional layers have 32 output channels, a kernel size of 3, a stride of 1, and a padding of 1. Due to limitations on the size of our ResNet, this is stronger in practice.

**Full ResNet:**  A full ResNet18 with the usual specifications.

**MNIST:**  The MNIST dataset contains 70k 28x28 greyscale images of hand-drawn digits in 10 classes (Lecun et al., 1998). We conduct our experiments with 49k training images and 21k test images. The classes are mutually exclusive. We use this dataset in Sec. 5.

**FashionMNIST:**  The FashionMNIST (FMNIST) dataset contains 70k 28x28 greyscale images of clothing in 10 classes (Xiao et al., 2017). We conduct our experiments with 49k training images and 21k test images. The classes are mutually exclusive. We use this dataset in Sec. 5.

**Kuzushiji-MNIST:**  The Kuzushiji-MNIST (KMNIST) dataset contains 70k 28x28 greyscale images of Japanese kanji in 10 classes (Clanuwat et al., 2018). We conduct our experiments with 49k training images and 21k test images. The classes are mutually exclusive. We use this dataset in Sec. 5.

**CIFAR10:**  The CIFAR10 dataset consists of 60,000 32x32 color images in 10 classes. The classes are mutually exclusive and include airplanes, automobiles, birds, cats, deer, dogs, frogs, horses, ships, and trucks (Krizhevsky et al., 2009).

**CIFAR5:** The CIFAR10 dataset, except without the classes of cat, bird, deer, dog and frog. We use this dataset in Sec. 5.

**CIFAR-100:** The CIFAR-100 dataset is similar to CIFAR-10 but contains 100 classes, each with 600 images, making a total of 60,000 32x32 color images. The 100 classes are grouped into 20 superclasses, and each image comes with a "fine" label (the class to which it belongs) and a "coarse" label (the superclass to which it belongs) (Krizhevsky et al., 2009). We use this dataset only for the Assumption 2 experiment in Appendix A.

**ImageNet:** The ImageNet dataset contains over 14 million annotated images organized according to the WordNet hierarchy. It is used in the ImageNet Large Scale Visual Recognition Challenge (ILSVRC) and includes a wide variety of object categories. The dataset provides both image-level annotations (indicating the presence or absence of an object class) and object-level annotations (bounding boxes around objects). ImageNet is a benchmark for image classification and object detection tasks (Deng et al., 2009). We use the smaller ImageNet used in ILSVRC. We use this dataset only for the Assumption 2 experiment in Appendix A.

E.3   BASELINE DETAILS

We provide an upper bound on the number of instances we can verify/falsify for experiment paradigms a) and b) respectively with k-Nearest Neighbors (Fix, 1985). Specifically, the regulator obtains the embeddings of the test or train set with a pretrained ResNet50 over ImageNet, and then verifies if the nearest neighbor to an object set instance is more than $\delta$ away, falsifying otherwise. This provides a baseline without the regulator having the entire test or train set at hand, but is of course susceptible to the data controller hiding the true test or train set.

Additionally, we use Athena (Sommer et al., 2022) to provide a baseline unlearning verification algorithm. Athena is a hypothesis test where the null hypothesis is verifying unlearning and the alternative hypothesis is falsifying unlearning. We poison $5\%$ of the training data with a backdoor trigger. Specifically, we set four random pixels to be 1 in the vectorized image and have the target class be 0 for all object set instances. In our main paper, the experiments are done such that the model is trained with additionally perturbed PGD object set instances, and Athena verifies over the original object set instances. This mimics the case in which the data controller keeps an adversarially perturbed version of $x_u$ in the train set.

Sommer et al. (2022) estimate a test statistic $\hat{r}$ based on backdoor accuracy over the object set and $f_m$. Additionally, they estimate parameter $q$ as the backdoor accuracy given the data controller has deleted the data. Then, one rejects the null hypothesis if $\hat{r} > t$ for a threshold $t \in [0, 1]$. The Type I error rate $\alpha$ of the hypothesis test can be written as follows:

**Proposition 3** *(Adapted from Sommer et al. (2022)) Suppose we have Type 1 error probability $\alpha$, parameter $q$, $n$ backdoored object set samples, and null hypothesis $H_0$ being compliance with partial data withdrawal. Suppose we have an estimate $\hat{r}$ such that if $\hat{r} > t$ we reject $H_0$. Then, for a given threshold $t \in [0, 1]$:*

$$\alpha_q^t = \Pr[\hat{r} > t | H_0, n] = \sum_{k > nt} \binom{n}{k} q^k (1-q)^{n-k} \tag{3}$$

We find such a $t$ by noting the following:

**Proposition 4** *Suppose Eq. (3) holds. Fix a binomial random variable $K \sim Bin(n, q)$. Then, $\alpha_q^t$ is equivalent to $\Pr[K > nt]$*

To demonstrate this, suppose Eq. (3) holds and fix a binomial random variable $K \sim Bin(n, q)$. Then, the cumulative distribution function of $K$ is given as $\Pr[K \leq m] = \sum_{k=0}^{m} \binom{n}{k} q^k (1-q)^{n-k}$. In particular, $\Pr[K \leq nt] = \sum_{k=0}^{nt} \binom{n}{k} q^k (1-q)^{n-k} = \sum_{k \leq nt} \binom{n}{k} q^k (1-q)^{n-k}$. It follows that $\Pr[K > nt] = 1 - \Pr[K \leq nt] = \sum_{k > nt} \binom{n}{k} q^k (1-q)^{n-k}$, and thus we are done.

In particular, since Eq. (3) is equivalent to $\Pr[K \leq nt] = 1 - \Pr[K > nt]$ we can efficiently numerically approximate the optimal $t$ given all the premises of proposition 4 to a sufficiently small

error rate $\gamma$, which is precisely what we do; we choose $\gamma = 1e - 6$ and use a simple bisection method.

Per the specifications of Sommer et al. (2022), we estimate $\hat{p}$ by training a model over poisoned data without deletion and computing backdoor accuracy. We then estimate $\hat{q}$ by training without the object set computing backdoor accuracy. Finally, we estimate $\hat{r}$ as the backdoor accuracy over the server's model given poisoned object set instances; in experiment paradigm b)'s case, the server's model retains additionally perturbed PGD object set instances. We also estimate the power of Athena, given $\hat{p}$ and $\hat{q}$, as follows:

**Proposition 5** *(Theorem 1 of Sommer et al. (2022)) Suppose all the premises of proposition 4 hold, additionally supposing parameter $p$. Then, the power or deletion confidence $\rho$ of Athena is given as follows:*

$$\rho = 1 - \sum_{k=0}^{n} \binom{n}{k} p^k (1-p)^{n-k} \cdot H[\sum_{l=0}^{k} \binom{n}{l} q^l (1-q)^{n-l} \leq 1 - \alpha] \qquad (4)$$

*where $H(\cdot)$ is the Heaviside step function such that $H(x) = 1$ if $x$ is True and $H(x) = 0$ otherwise.*

Per standard statistical notation, given that $\beta$ is the Type II error probability, $\rho = 1 - \beta$.

### E.4 EXPERIMENT SPECIFICATIONS

We train all networks with 200 epochs with minibatch stochastic gradient descent. In particular, we choose a momentum of 0.9 and a weight decay of $1 * 10^{-4}$. Throughout, we use cross entropy loss. We also use a multi-step learning rate scheduler, starting at 0.1 with 0.1 decay. We use a batch size of 128. We use standard data for our CIFAR5 experiments, including random horizontal flips and random cropping.

Datasets in paradigm b) are downsampled to 10000 instances such that we reach 0 or approximately 0 training loss. We also sample two object set instances per class for all experiments. We also choose $\delta = \frac{1}{510}$ throughout, providing an ablation study on this towards the end of our main paper. Finally, we use the transformations as stated in the paper for MNIST, KMNIST, and FMNIST. However, for CIFAR5–as we demonstrate in Appendix B.2–a principled alteration to the object set is required. We provide an ablation study on how our experiments are affected when using this transformations across all of MNIST in Appendix B.2. To be clear, the special transformation used for CIFAR5 is given as follows:

a) First, we train the model we verify over including the object set.

b) Then, we perform a PGD attack with $\epsilon = 0.1$, $\alpha = \frac{2}{255}$, and 100 steps on the object set instances.

c) Finally, we apply this PGD noise to the object set instances.

Notably, this PGD noise is only optimized once for a batch of object set samples. We also chose $\epsilon$ sufficiently large such that it effectively induced misclassification but sufficiently small such that it still made the images itself interpretable with respect to the ground truth.

## F KEYWORD AND SYMBOL TABLE

The below keyword and symbol table provides definitions for key concepts and mathematical symbols and used throughout the paper. On the left hand side, the keyword or symbol is presented. On the right hand side, its definition is presented.

| Keywords | |
|---|---|
| "data controller" | The provider of machine learning models. |
| "user" | The user of the data controller's service, which has an underlying machine learning model processing data to make predictions. |
| "GDPR" | The European Union's General Data Protection Regulation. |
| "right to be forgotten" | The right stipulated in the GDPR such that if a user invokes it, a data controller must delete that user's data. This data deletion is not restricted to just the training data. |
| "machine unlearning" | A field of research devoted to developing algorithms to make complying with the right to be forgotten easier for data controller. |
| "right to object" | The right stipulated in the GDPR such that if a user invokes it, a data controller must cease processing that user's data. This can be invoked at both train and test time. |
| "partial data withdrawal" | When a user requests that their data be deleted from subset of the processing e.g. training. |
| "total data withdrawal" | When a user requests that their data be deleted from all processing e.g. both training and testing. |
| "auditable compliance" | Compliance with a legal policy such that an unbiased third party can verify said compliance. |
| "soundness" | The property of a verification algorithm such that whenever it verifies, it does so correctly. |
| "completeness" | The property of a verification algorithm such that whenever it falsifies, it does so correctly. |

**Symbols**

| | |
|---|---|
| $\forall$ | The logical quantifier "for all". |
| $\exists$ | The logical quantifier "there exists". |
| $x \in X$ | Read as "the element $x$ in set $X$". |
| $u$ | User $u$ who invokes total data withdrawal. |
| $T$ | The task $T$ for which total data withdrawal is invoked over. |
| $[n]$ | The set $\{1, ..., n\}$. $[d]$ and $[o]$ are used throughout. |
| $\boldsymbol{x}_u$ | A real vector belonging to user $u$. |
| $\mathbb{R}^d$ | A real vector space of dimension $d$. |
| $\mathbb{R}^o$ | A real vector space of dimension $o$. |
| $[0,1]^d$ | A subset of $\mathbb{R}^d$ such that all vectors contained have entries between 0 and 1. |
| $\|\boldsymbol{x}\|_\infty$ | The $\ell_\infty$ norm of a real vector $\boldsymbol{x}$. Supposing that $\boldsymbol{x} \in \mathbb{R}^d$, this evaluates to $\max_{i \in [d]} \|x_i\|$ where $\max_{x \in X}$ returns the maximum over all elements in $X$. |
| $\mathcal{X}$ | Sample space. Taken to be a subset of $[0,1]^d$ throughout. |
| $\mathcal{Y}$ | Label space. Taken to be a subset of $\mathbb{R}^o$ throughout. |
| $f_0$ | The neural network used by the data controller prior to total data withdrawal being invoked. $f_0$ is a function between $[0,1]^d$ and $\mathbb{R}^o$. |
| $T$ | Task $T$ is the task that the user has invoked total data withdrawal over. |
| $S_{\text{tr}}$ | The training data of the form $\{(\boldsymbol{x}_i, \boldsymbol{y}_i)\}_{i=1}^{N_{\text{tr}}}$. That is, it is a set of cardinality $N_{\text{tr}}$ containing tuples sampled from $\mathcal{X} \times \mathcal{Y}$, where $\times$ denotes the Cartesian product. |
| $X_{\text{tr}}$ | The train set instances, in particular the set containing the first elements in $S_{\text{tr}}$. A subset of $\mathcal{X}$. |
| $S_{\text{te}}$ | The test data of the form $\{(\boldsymbol{x}_i, \boldsymbol{y}_i)\}_{i=1}^{N_{\text{te}}}$. That is, it is a set of cardinality $N_{\text{te}}$ containing tuples sampled from $\mathcal{X} \times \mathcal{Y}$, where $\times$ denotes the Cartesian product. |
| $X_{\text{te}}$ | The test set instances, in particular the set containing the first elements in $S_{\text{te}}$. A subset of $\mathcal{X}$. |
| $f(\boldsymbol{x})_i$ | The $i$th output logit for a neural network $f$. $f$ is used throughout as either $f_0$ or $f_m$. The classification of $\boldsymbol{x}$ by $f$ is thus given as $t = \max_{i \in [o]} f(\boldsymbol{x})_i$. |
| $\bar{B}_\delta(\boldsymbol{x}_u)$ | The closed $\delta$-ball around $\boldsymbol{x}_u$ defined as $\bar{B}_\delta(\boldsymbol{x}_u) = \{\boldsymbol{x} \in [0,1]^d : \|\boldsymbol{x} - \boldsymbol{x}_u\|_\infty \leq \delta\}$, using standard set builder notation. |
| $f_m$ | The potentially altered neural network used in place of $f_0$ after the user $u$ invokes total data withdrawal over task $T$. |
| $|\mathcal{Y}|$ | The cardinality of the label set. |
| $\epsilon$ | A real scalar satisfying $\forall \boldsymbol{x}_n, \boldsymbol{x}_m \in X_{\text{tr}}, X_{\text{te}}, \|\boldsymbol{x}_n - \boldsymbol{x}_n\|_\infty > \epsilon$ for $m \geq n$. |
| $\Longleftrightarrow$ | Used to denote if and only if, or a condition being necessary and sufficient for another condition. |
| $\Longleftarrow$ | When demonstrating $P \iff Q$, in our paper this refers to the part of the argument where we demonstrate $P \Longleftarrow Q$. |

| | |
|---|---|
| $\implies$ | When demonstrating $P \iff Q$, in our paper this refers to the part of the argument where we demonstrate $P \implies Q$. |
| $\gamma$ | Used to denote any member of $[o]$ that is not $t$. |
| $\sigma(\boldsymbol{x}) = \max\{0, \boldsymbol{x}\}$ | A ReLU activation function. |
| $h = g \circ f_m$ | The function $h$ composed of the functions $g$ and $f_m$. Used in proposition 2. |
| $X_{\text{tr}} \cup X_{\text{te}}$ | The set containing all instances in the training and test data. |
| $X_{\text{o}}$ | The object set, or the set of instances total data withdrawal has been invoked over for task $T$. A subset of $X_{\text{tr}} \cup X_{\text{te}}$. |
| $X_{\text{o}}^{\text{adv}}$ | The set of closed $\delta$-balls around the object set instances in $X_{\text{o}}$, formally defined as $\bigcup_{i=1}^{\|X_{\text{o}}\|} \bar{B}_\delta(\boldsymbol{x}_i)$ where $\boldsymbol{x}_i \in X_{\text{o}}$ and $\|X_{\text{o}}\|$ is the cardinality of $X_{\text{o}}$. |
| $\mathcal{X} \setminus X_o$ | The set of all instances in the sample space that are not in the object set. |
| $\mathcal{N}(0, 0.01)$ | A Gaussian distribution with mean $0$ and variance $0.01$. |

