# OpenReview forum: "Auditing Data Controller Compliance with Data Withdrawal"
_ICLR.cc/2025/Conference — ICLR 2025 Conference Withdrawn Submission_

### Official Review · Reviewer_vwsT · 2024-10-28

**Soundness:** 3
**Presentation:** 3
**Contribution:** 2
**Rating:** 3
**Confidence:** 5

**Summary:**

This paper seeks to audit if a particular data point was entirely unlearnt from a model -- essentially verifying that the model owner carried out the GDPR "right to be forgotten" requirement. It frames the problem as that of verifying that the resulting classifier (after unlearning) misclassifies the point x to be unlearnt as well as its entire $\delta$ neighborhood. It then uses an existing robustness verification algorithm due to Wang et al 2021 to do the verification, and provides some nice experimental evidence.

**Strengths:**

* The paper studies an important problem -- verifying and auditing if a model complied with GDPR is indeed a major technical  challenge in trustworthy ML that we are barely scratching the surface of. More work on this topic should be encouraged.

**Weaknesses:**

The major weakness of the paper is the framing of the problem. There are three assumptions made -- there are 3 or more labels, training data points are at most distance epsilon from each other, and the model $f_m$ obtained after removal has training loss going to zero (basically that it correctly classifies all training points).

I think Assumptions 1 and 2 are reasonable; in fact a version of Assumption 2 was empirically shown for a number of common datasets by [1]. Assumption 3 seems a little less reasonable to me, especially for the larger models of today where one can barely make a few passes over the training data.

In any case, the conclusion drawn from all three of these is that any model $f_m$ that is capable of verification MUST misclassify all points in a ball around the removed point. This is technically true -- if the points are not misclassified, then one cannot statistically verify removal.

However, this is not a sensible requirement of any model. Suppose the data point x is very much an "in-distribution" point -- an ordinary looking zero in MNIST -- then it is unreasonable to ensure that removing it misclassifies all zeros in the vicinity. In fact, it's quite likely that if x never even existed, and if we never trained the model with x, the model would have classified x correctly. If this is the case, then it should classify x correctly when it is removed as well -- otherwise we are hurting generalization properties of the model.

Another key word in the conclusion drawn about verification is "statistically" -- it is true that one cannot verify statistically, but it might still be possible to verify removal cryptographically without resorting to such strong assumptions. For example, there is a growing body of work on cryptographical verification of properties of models using tools such as one-way functions and zero-knowledge. I would encourage the authors to check out some of that body of work. [2, 3, 4, 5]

[1] Yang, Yao-Yuan, et al. "A closer look at accuracy vs. robustness." Advances in neural information processing systems 33 (2020): 8588-8601.

[2] https://arxiv.org/abs/2210.09126

[3] Garg, Sanjam, Shafi Goldwasser, and Prashant Nalini Vasudevan. "Formalizing data deletion in the context of the right to be forgotten." Annual International Conference on the Theory and Applications of Cryptographic Techniques. Cham: Springer International Publishing, 2020.

[4] Garg, Sanjam, et al. "Experimenting with zero-knowledge proofs of training." Proceedings of the 2023 ACM SIGSAC Conference on Computer and Communications Security. 2023.

[5] Yadav, Chhavi, et al. "FairProof: Confidential and Certifiable Fairness for Neural Networks." arXiv preprint arXiv:2402.12572 (2024).

**Questions:**

See Weaknesses.

---

### Official Review · Reviewer_r14D · 2024-11-03

**Soundness:** 2
**Presentation:** 2
**Contribution:** 1
**Rating:** 3
**Confidence:** 3

**Summary:**

This paper addresses the concept of total data withdrawal, where a user requests that their data be removed entirely from a machine learning system, affecting both data at training and test time. The authors develop a formal framework to audit data controllers’ compliance with these withdrawal requests, ensuring that user data is neither used to train the model nor make predictions at test time. To validate compliance, the authors propose a regulatory check in which the data controller must misclassify the user’s instance and all perturbed variations of it within a specified range, preventing potential circumventions through adversarial perturbations.

**Strengths:**

- The idea of verifying compliance through misclassification of instances within a closed ball is somewhat innovative and hopes to enforce more stringent guarantees against data misuse.
- The work introduces a structured approach to the issue of data withdrawal from both training and test data, which could benefit the evolving discussion around data privacy and user rights in machine learning.

**Weaknesses:**

- **Relevance of Total Data Deletion**: First, The core problem formulation - total data deletion - raises practical concerns. During the deployment phase, predictions are typically made on new data instances provided voluntarily by users, making it unclear why users would request data deletion yet still expect service predictions. If a user continues to provide data, making predictions would likely be legal, limiting the relevance of total data withdrawal to the training phase alone. Second, effective techniques for verifying data deletion from the training set already exist, such as membership inference frameworks that achieve reliable results in real-world settings (e.g., [1]). This casts doubt on the necessity of the proposed approach.

- **Infeasibility of Methodology**: The proposed verification method appears impractical, as it requires adversarial perturbation across all training data points to identify potential instances of non-compliance. This would be computationally prohibitive, particularly for large datasets, and could significantly degrade the model’s performance due to the extensive perturbations required for each training instance.

- **Unrealistic Assumptions**: Some assumptions in this work are prohibitively strict, notably the requirement of zero training error. In many models, especially large language models (LLMs), achieving zero training error is impractical due to their immense size and complexity. If this assumption is unmet, the authors' proof seems to fail, limiting the framework’s applicability to a small subset of ML models that can achieve such high accuracy on the training data.


**Suggestions for improvement**

- To address the issue of practical relevance, the authors should consider focusing exclusively on the training phase in the context of data deletion. This approach could help streamline the method and align it with established deletion verification practices.
- The feasibility of the proposed approach could be improved by considering alternative compliance verification techniques that don’t rely on adversarial perturbation of all training data points.
- Re-evaluating the assumptions, especially zero training error, would make the approach more applicable to modern machine learning models and increase its potential impact.


----
**References**

[1] Gaussian Membership Inference Privacy, https://proceedings.neurips.cc/paper_files/paper/2023/hash/e9df36b21ff4ee211a8b71ee8b7e9f57-Abstract-Conference.html

**Questions:**

- Could the authors clarify their motivation for considering both training and test time deletion? Are specific use cases or regulatory requirements they had in mind that necessitate verification of deletion at test time as well?

- Could the authors provide complexity analysis or empirical runtime measurements as well as performance evaluation for their method on different dataset size? This would allow a more concrete understanding of scalability limitations.

- Could the authors discuss how their method might be adapted or relaxed to handle models that don't achieve zero training error?

---

### Official Review · Reviewer_5nKQ · 2024-11-04

**Soundness:** 1
**Presentation:** 1
**Contribution:** 1
**Rating:** 1
**Confidence:** 4

**Summary:**

The paper focuses on the problem of machine unlearning verification. The author defines a set assumption and showcases that under that set of assumptions their approach can be effective.
They also compare their approach to one of previous works and show they can outperform the existing approach.

**Strengths:**

Very important problem and paper does a good job of motivating the problem.

**Weaknesses:**

In section 3 the authors claim that an approach is verifiable if and only if the model misclassified the points that are selected to be withdrawn. This can be incorrect given a different model. Let's assume a SVM that is trained and the model trainer will release every step of the training with zero knowledge proof of inputs and outputs or some other trust mechanism approach (trusted hardware, …). Now if you remove a point which is not around the decision boundaries it won't change anything about the function so it won't misclassify those points as results you are claiming this data is not auditable withdrawn, however, this is not true and an auditor can easily verify this. This makes most of the framing in this work not applicable in most applications.



Moreover the authors also mention that a point has to be verifiably withdrawn from the training data if the point should be wrong on all points around a point. Again this can be very problematic as we know adversarial examples exist and if the ball around the point is not very small given a model we can adversarially construct a point such that it can be classified correctly and also actually verifying this in practice can be very hard. As to ensure such a point does not exist we have to do an exhaustive search.

Similar to the previous point we can also construct adversarial scenarios where even if a model answers wrong on a point and all surrounding points it does not mean it is verifiable and not used in the training. Just as an example assume I train a model on many points and when I want to remove a point I just add a if statement that if I get any point near those selected points return a wrong label, the paper here considers this verifiable withdrawn, however, this is falsified. In general the goal of designing an approach for auditing data removal is to make sure the approach does not have high false positives and cannot be easily bypassed, which is not the case for this approach.


The 3rd assumption in Section 3 is "The training loss of f_m goes to 0", I am not sure what this exactly means. Does it go in the limit or it is zero when released ?


The experiments in this work are very unrealistic. Significantly modifying the points will be easily detectable, however, this is not going to in practical scenarios. If the authors want, they can evaluate many face recognition datasets as suggested by themselves.

**Questions:**

Mentioned above

---

### Official Review · Reviewer_SEbt · 2024-11-05

**Soundness:** 2
**Presentation:** 2
**Contribution:** 2
**Rating:** 3
**Confidence:** 2

**Summary:**

The paper focuses on auditing the removal of a certain user's data from both the training and test sets of a model (which they call total data removal). This involves proving unlearning (when it happens) and falsifying it (when unlearning does not happen). The proposed framework relies on delivering provable guarantees that all perturbed examples around the current one are incorrectly classified, and is supported by experimental results.

**Strengths:**

The paper identifies (and attempts to formalize) a very important problem. Verifying unlearning has already been considered but provably refuting compliance with data deletion requests is just as important, as this paper points out.

**Weaknesses:**

**Setting not fully justified**: I find the requirement that the model misclassify all instances in a ball around the original instance to be counterintuitive.
- If a model generalizes well to an unseen data point, this is still considered a violation as per the proposed framework. This does not map to practical use cases in my head machine learning is fundamentally based on generalization. How can you justify this seemingly extreme requirement?
- In particular, this seems to be the reason that the examples that need to be deleted are highly out of distribution (see line 325, for instance). It is still useful to study methods that can only verify the removal of out-of-distribution examples, but this has to be stated upfront.
- Further, it seems to me that this definition of data removal can be gamed. It is very easy to misclassify an example by simply changing its label. This does not mean the data has been removed from training. Membership inference attacks might still be able to detect the presence of the example. How can you justify this serious drawback?

**Mathematical rigor**: The paper talks about establishing a necessary and sufficient condition. There is also what looks like a proof sketch on page 5. However, there are no mathematical statements (theorems, lemmas, etc.) that clearly state the assumptions and the conclusion. Moreover, there is no fully rigorous proof either (only a proof sketch).

**Unclear algorithm/complexity**: The paper proposes to use CROWN BaB, which is not described. The entire algorithm looks like a black box for this reason. It is helpful to give a high level idea for those unfamiliar with this algorithm (as it is not a textbook algorithm).  Further, what are the time and space complexity of this method? Also, how to choose the parameter $\delta$?

**Minor comments:**
   - Figure 1 is confusing. What is it supposed to convey?
   - Eq. 1: is it argmax instead of max?
   - Experiments: baselines are not explained clearly. What are the baselines trying to achieve? Why are they reasonable to use here?

**Questions:**

- Some failures are reported in line 362: what are the failure modes and why do they happen? Also, don't failures defeat the purpose of auditing?
- The paper describes auditing the deletion of test data to be highly important. How do you actually verify this?

---

### Author Response · Authors · 2024-11-13

Dear reviewers,

Thank you for your thoughtful comments. We have decided to withdraw the paper at this time; we will incorporate your feedback and resubmit at another venue.

---

### Note · Authors · 2024-11-13

I have read and agree with the venue's withdrawal policy on behalf of myself and my co-authors.